# Repurposing of the Cardiovascular Drug Statin for the Treatment of Cancers: Efficacy of Statin–Dipyridamole Combination Treatment in Melanoma Cell Lines

**DOI:** 10.3390/biomedicines12030698

**Published:** 2024-03-21

**Authors:** Nanami Irie, Kana Mizoguchi, Tomoko Warita, Mirai Nakano, Kasuga Sasaki, Jiro Tashiro, Tomohiro Osaki, Takuro Ishikawa, Zoltán N. Oltvai, Katsuhiko Warita

**Affiliations:** 1Graduate School of Science and Technology, Kwansei Gakuin University, 1 Gakuen Uegahara, Sanda 669-1330, Japan; 2Department of Biomedical Sciences, School of Biological and Environmental Sciences, Kwansei Gakuin University, 1 Gakuen Uegahara, Sanda 669-1330, Japan; 3Department of Veterinary Anatomy, School of Veterinary Medicine, Tottori University, 4-101 Koyama Minami, Tottori 680-8553, Japan; 4Department of Veterinary Clinical Medicine, School of Veterinary Medicine, Tottori University, 4-101 Koyama Minami, Tottori 680-8553, Japan; 5Department of Anatomy, School of Medicine, Aichi Medical University, 1-1 Yazakokarimata, Nagakute 480-1195, Japan; 6Department of Pathology and Laboratory Medicine, University of Rochester, 601 Elmwood Ave, Rochester, NY 14642, USA; zoltan_oltvai@urmc.rochester.edu

**Keywords:** statin, drug repositioning, carcinostatic effect, mevalonate pathway, HMG-CoA reductase, dipyridamole, combination therapy, human melanoma, spontaneously occurring canine melanoma, comparative oncology

## Abstract

Metastatic melanoma has a very poor prognosis. Statins, 3-hydroxy-3-methyl-glutaryl-coenzyme A reductase (HMGCR) inhibitors, are cholesterol-lowering agents with a potential for cancer treatment. The inhibition of HMGCR by statins, however, induces feedback, which paradoxically upregulates HMGCR expression via sterol regulatory element-binding protein-2 (SREBP2). Dipyridamole, an antiplatelet agent, is known to inhibit SREBP2 upregulation. We aimed to demonstrate the efficacy of statin–dipyridamole combination treatment in both human and spontaneously occurring canine melanoma cell lines. The half maximal inhibitory concentration (IC_50_) of atorvastatin showed a 68–92% reduction when combined with dipyridamole, compared with that of atorvastatin alone. In some melanoma cell lines, cell proliferation was suppressed to almost zero by the combination treatment (≥3 μM atorvastatin). Finally, the BRAF inhibitor, vemurafenib, further potentiated the effects of the combined statin–dipyridamole treatment in BRAF V600E mutation-bearing human melanoma cell lines. In conclusion, the inexpensive and frequently prescribed statin–dipyridamole combination therapy may lead to new developments in the treatment of melanoma and may potentiate the effects of vemurafenib for the targeted therapy of BRAF V600E-mutation bearing melanoma patients. The concordance between the data from canine and human melanoma cell lines reinforces this possibility.

## 1. Introduction

Melanoma is an aggressive skin tumor responsible for most skin-cancer-related deaths globally [1]. There is a large interracial difference in the occurrence of melanoma, with the highest incidence in White populations (29.7 males and 19.1 females per 100,000), followed by Hispanics (4.4 males and 4.7 females per 100,000), Asians, and Black populations (1.1 males and 1.0 females per 100,000) [2]. In Japan, melanoma is a rare cancer usually diagnosed at a late stage compared to the time of diagnosis in Western countries [3]. Currently, the therapeutic options for melanoma include surgical resection, chemotherapy, radiotherapy, photodynamic therapy, immunotherapy, and targeted therapy [1]. Immune checkpoint inhibitor therapy targeting CTLA-4 or programmed cell death protein 1 (PD-1) and its ligand (PD-L1) and targeted therapy with kinase inhibitors (BRAF and mitogen-activated extracellular signal-regulated kinase (MEK) inhibitors) are the current standard treatments for melanoma [4]. However, there are two challenges: adverse events that can lead to skin and gastrointestinal toxicity, and reduced efficiency due to resistance to immune, chemo-, and targeted therapies [1]. This necessitates the search for additional therapy options.

Compactin (mevastatin), the first statin, was discovered by Dr. Akira Endo in Japan from the blue-green mold *Penicillium citrinum* Pen-51, isolated from a rice sample collected at a grain shop in Kyoto [5,6]. Thereafter, other statins (compactin analogs) were developed and are now used worldwide to treat hyperlipidemia. Statins function as 3-hydroxy-3-methyl-glutaryl-coenzyme A reductase (HMGCR) inhibitors and suppress cholesterol biosynthesis by specifically inhibiting HMGCR, the rate-limiting enzyme of the mevalonate pathway. This metabolic pathway is responsible for the synthesis of cholesterol and other important biomolecules, such as dolichols, which are crucial for glycosylation; ubiquinone, crucial for mitochondrial electron transport processes; and farnesyl pyrophosphate (FPP) and geranylgeranyl pyrophosphate (GGPP), which are essential for protein prenylation [7,8]. Therefore, the depletion of mevalonate pathway intermediates by statins interferes with the post-translational modification and activation of small guanosine triphosphatases (GTPases) (such as Ras, Rho, or Rac) and their downstream signaling, which affects cell proliferation and differentiation or leads to apoptosis [7]. Additionally, the mevalonate pathway supports tumorigenesis and is deregulated in human cancers [8]. Moreover, statins have been reported to have anticancer effects in different types of cancers, including lung, kidney, bladder, colon, breast, and prostate cancer [7,9,10,11,12]. Therefore, there is substantial interest in repurposing statins as therapeutic agents for cancer. Moreover, Jiang et al. suggested that because of their efficacy, statins may address the shortcomings of conventional cancer treatments and should be considered in the context of combined therapies for cancer [7].

However, the inhibition of HMGCR activity by statins paradoxically induces a feedback regulation in statin-treated cells, eventually resulting in the upregulation of HMGCR expression. The underlying mechanism involves the transcription of *HMGCR*, which is activated by sterol regulatory element binding protein-2 (SREBP2) [8]. As this homeostatic mechanism weakens the effectiveness of statins and creates the need to use higher doses for cancer inhibition, statins have yet to be successfully repurposed as cancer therapeutic drugs. Thus, for the successful use of statins in cancer therapy, the prevention of the homeostatic feedback loop that blunts statin efficacy may be required [13].

Our previous studies showed that atorvastatin is effective against all melanoma cells in both human [14] and spontaneously occurring canine melanoma cell lines [15,16]. Furthermore, plasma levels of atorvastatin, comparable to those used for the treatment of hypercholesterolemia, have been reported to inhibit the metastasis of human melanoma cells overexpressing RhoC in a xenograft model [17]. In addition, simvastatin and fluvastatin have been reported to inhibit tumor metastasis and growth by suppressing the Rho signaling pathway and to markedly improve the survival rate in a mouse melanoma model [18]. These findings highlight the need for further investigations of statins toward clinical applications in the treatment of metastatic melanoma.

In this study, we aimed to demonstrate that dipyridamole, an antiplatelet agent, potentiates the anticancer effects of statins against both human and spontaneously occurring canine melanoma by interfering with the statin-induced activation of SREBP2-driven HMGCR upregulation. Both statins and dipyridamole are currently approved drugs; therefore, our findings may contribute to the immediate advancement of their evaluation in clinical trials.

## 2. Materials and Methods

### 2.1. Cell Culture

Melanoma-derived SK-MEL-5 and MDA-MB-435 cells were obtained from the Division of Cancer Treatment and Diagnosis (DCTD) Tumor Repository (National Cancer Institute, Frederick, MD, USA). Three canine melanoma primary tumor cell lines, YCC, ITP, and HTR, established by Osaki et al. [19], were also used in this study. They tested negative for *Mycoplasma* using an e-Myco^TM^ Mycoplasma PCR Detection Kit (iNtRON Biotechnology, Sungnam, Republic of Korea). The cell lines were cultured in RPMI 1640 medium (Thermo Fisher Scientific, Waltham, MA, USA) supplemented with 10% heat-inactivated fetal bovine serum (Biosera, Boussens, France) and penicillin/streptomycin (Fujifilm Wako Pure Chemical, Osaka, Japan; final concentration of 100 units/mL penicillin G and 100 μg/mL streptomycin) in a humidified incubator at 37 °C with 5% CO_2_.

### 2.2. Reagents

Atorvastatin (Sigma-Aldrich, St. Louis, MO, USA) and dipyridamole (Sigma-Aldrich) were dissolved in dimethyl sulfoxide (DMSO; Fujifilm Wako Pure Chemical; final concentration of 0.1% in RPMI 1640 medium) at final concentrations of 0.3–30 μM and 1–100 μM, respectively. Vemurafenib, dissolved in 0.1% DMSO (Selleck Chemicals, Houston, TX, USA), was added to RPMI at final concentrations of 0.1–10 μM. DMSO was added to the control groups as the vehicle control at final concentrations of 0.2% and 0.3% in atorvastatin–dipyridamole combination treatment and atorvastatin–dipyridamole–vemurafenib combination treatments, respectively. Geranylgeranyl pyrophosphate ammonium salt solution (Sigma-Aldrich) was used at a final concentration of 20 μM with reference to the paper by Zhou et al. [20]. Methanol (final concentration, 1%; Fujifilm Wako Pure Chemical) was added to the control groups as the vehicle control; 0.3% DMSO and 1% methanol have no cytotoxic effects on human cancer cell lines [21]. All cell culture experiments were performed in triplicate.

### 2.3. Morphological Analysis by Optical and Fluorescence Microscopy

The cells were seeded in 6-well plates at a density of 1 × 10^5^ cells/mL, incubated overnight prior to treatment with 3 µM dipyridamole, 10 µM atorvastatin, or their combination for 72 h, and then imaged using an Olympus IX71 microscope (Olympus, Tokyo, Japan).

Cultured cells were fixed with 2% paraformaldehyde (Nacalai Tesque, Kyoto, Japan) for 30 min and washed with phosphate-buffered saline (PBS). They were then permeabilized with 0.1% Triton-X-100 (Nacalai Tesque) for 15 min. After washing with PBS, the cells were treated with 2% bovine serum albumin for 15 min to block any non-specific proteins. Subsequently, 100 nM Acti-stain^TM^ 488 phalloidin (PHDG1; Cytoskeleton, Denver, CO, USA) was used for F-actin staining. Following a PBS wash, the cells were subjected to nuclear staining using 5 µg/mL Hoechst 33342 (Dojindo, Kumamoto, Japan) for 15 min and mounted using Fluoromount/Plus™ (Diagnostic BioSystems, Pleasanton, CA, USA). Images were captured using a digital fluorescence microscope (BZ-X810; Keyence, Osaka, Japan).

### 2.4. Cell Proliferation Assay

The cells were seeded in 96-well plates at a density of 2.5 × 10^4^ cells/mL and incubated overnight, prior to treatment with each reagent, for 72 h. Five different assays were performed: (1) single treatment with each concentration of dipyridamole, (2) treatment with each concentration of atorvastatin with (or without) 3 µM dipyridamole, (3) rescue experiment by GGPP against 10 µM atorvastatin and 3 µM dipyridamole combination treatment, (4) single treatment with each concentration of vemurafenib, and (5) combined treatment with 3 µM atorvastatin, 3 µM dipyridamole, and 0.1 or 0.3 µM vemurafenib. Absorbance was measured at 450 nm using an 800 TS microplate reader (BioTek, Winooski, VT, USA) and a Cell Counting Kit-8 (CCK-8, Dojindo). The IC_50_ values were calculated by constructing sigmoid curves based on cell viability using ImageJ software version 1.52a (National Institutes of Health, Bethesda, MD, USA).

### 2.5. RNA Extraction and Quantitative Real-Time Polymerase Chain Reaction (qRT-PCR)

Cells were seeded in 6-well plates and cultured until subconfluence. An ISOSPIN Cell and Tissue RNA kit (Nippon Gene, Tokyo, Japan) was used to extract total RNA for qRT-PCR from the cells 24 h after the addition of 1 µM and 10 µM atorvastatin and/or 3 µM dipyridamole in accordance with the manufacturer’s instructions.

The RNA samples were treated with RNase-free DNase to eliminate genomic DNA contamination. cDNA was synthesized from 1 μg of RNA using the ReverTra Ace^®^ qPCR RT Master Mix kit (Toyobo, Osaka, Japan). Expression levels of *HMGCR* were examined. The canine primer sets used for gene expression analysis were as follows: *HMGCR* forward 5′-GGAGAGCCTCTGAGTGGTTG-3′ and reverse 5′-TGTTCACTGCCACTTCCGTG-3′; *GAPDH* (internal reference) forward 5′-GGTAGTGAAGCAGGCATCGG-3′ and reverse 5′- TTACTCCTTGGAGGCCATGTG-3′. PCR was conducted using Applied Biosystems PowerUp SYBR Green Master Mix and a QuantStudio^®^ 3 Real-Time PCR System (Thermo Fisher Scientific). The primer sets for humans and the PCR conditions have been described previously [20].

### 2.6. Western Blotting

The cells were seeded in 6-well plates and cultured until reaching subconfluence. Protein was extracted after 24 h of the addition of 10 µM atorvastatin, 3 µM dipyridamole, the drug combination, or 0.2% DMSO.

Western blotting was performed as described previously [20]. HMGCR protein levels were determined using an anti-HMGCR mouse monoclonal antibody (1:1000 dilution; AMAb90618; Atlas, Cambridge, UK). Anti-glyceraldehyde-3-phosphate dehydrogenase (GAPDH; 14C10) rabbit monoclonal antibody (1:1000 dilution, #2118; Cell Signaling Technology, Beverly, MA, USA) was used to detect GAPDH as an internal standard.

### 2.7. SEM Sample Preparation and Imaging

Cells treated with 5 μM atorvastatin alone or 3 μM dipyridamole in combination were imaged using SEM, as described below. Cells were cultured on plastic disks (Cell Disk LF1, Sumitomo Bakelite, Tokyo, Japan) and washed once with warm PBS. After aspirating the wash solution, cells were fixed with 2.5% glutaraldehyde diluted in PHEM buffer (pH 7.4; 60 mM PIPES, 25 mM HEPES, 10 mM EGTA, and 2 mM MgCl_2_ in H_2_O) at 25 °C for 30 min. Then, cells were washed with warm PHEM buffer for 5 min thrice, followed by post-fixation with 1% osmium solution dissolved in PHEM buffer for 30 min at 25 °C. After washing thrice with warm PHEM buffer for 5 min, the samples were dehydrated in an ascending ethanol series. Then, ethanol was replaced with 100% t-butyl alcohol, and the cell disks immersed in t-butyl alcohol were kept in the refrigerator (4 °C). Subsequently, the cells on the cell disks were lyophilized using a VFD-21 t-butanol freeze dryer (Vacuum Device, Ibaragi, Japan) and then coated with platinum/palladium (Pt/Pd) using a HITACHI E-1045 ion sputter coater (Hitachi High-Technologies, Tokyo, Japan). The cell surfaces were observed using a HITACHI SEM SU-8020 (Hitachi High-Technologies) at an accelerating voltage of 5 kV.

### 2.8. Statistical Analysis

Statistical analyses were performed using the Microsoft Excel add-in software (Bell Curve for Excel, version 4.02; Social Survey Research Information, Tokyo, Japan). Dunnett’s test or a Tukey–Kramer post hoc test were used to compare cell viability and *HMGCR* mRNA expression levels in each atorvastatin-treated or combination atorvastatin- and dipyridamole-treated group with those of the controls. Statistical significance was set at *p* value < 0.05. All the experiments were repeated at least three times, independently.

## 3. Results

### 3.1. Dipyridamole Co-Treatment Augments the Cytostatic Effect of Atorvastatin in Both Human and Canine Melanoma Cell Lines

Regarding cell morphology after drug treatments, atorvastatin treatment alone induced cellular shrinkage and apoptosis-like morphological changes both in human and canine melanoma cell lines compared with those in sham-treated cells (Appendix A). However, we observed no effect on cell morphology when cells were treated with dipyridamole alone (Appendix A). A combined treatment with atorvastatin and dipyridamole induced more pronounced morphological changes than atorvastatin treatment alone (Appendix A).

### 3.2. Half-Maximal Inhibitory Concentration (IC_50_) Value of Dipyridamole Exceeds 16.85 μM

In the human melanoma cell lines MDA-MB-435 and SK-MEL-5, and canine melanoma cell line ITP, more than 10 μM dipyridamole had an inhibitory effect on cell proliferation (Appendix A). In the canine melanoma cell lines HTR and YCC, dipyridamole treatment increased cell proliferation at lower concentrations; however, at higher concentrations, cell proliferation was inhibited (Appendix A). IC_50_ values of dipyridamole for MDA-MD-435, SK-MEL-5, HTR, ITP, and YCC were 40.31, 29.77, 19.75, 16.85, and 29.81 μM, respectively (Appendix A).

### 3.3. Combined Atorvastatin and Dipyridamole Treatment Enhances the Cytostatic Effect in Both Human and Canine Melanoma Cell Lines Compared with Atorvastatin Treatment Alone

A significant decrease in cell proliferation was observed in the combination treatment group at lower concentrations of atorvastatin compared with that in the atorvastatin monotherapy group (Figure 1).

The addition of dipyridamole reduced the concentration of atorvastatin required to significantly inhibit the cell proliferation of MDA-MB-435 cells, from 10 μM to 1 μM (1/10-fold), and of SK-MEL-5 cells from 10 μM to 3 μM (about 1/3-fold). In HTR cells, atorvastatin concentration reduced from 1 μM to 0.3 μM (about 1/3-fold), and in ITP and YCC cells from 3 μM to 1 μM (1/3-fold). The effects of atorvastatin alone or a combination of atorvastatin and dipyridamole depended on the cancer cell type. Notably, the addition of low concentrations of the drug increased the proliferation of some cells. In canine YCC cells, the combination of 3–30 µM atorvastatin and dipyridamole reduced cell proliferation to 0, while 30 µM atorvastatin alone reduced cell proliferation to 0. In ITP cells, the combination of 30 µM atorvastatin and dipyridamole reduced cell proliferation to zero. The IC_50_ values are shown in Table 1 and Appendix A.

### 3.4. Statin and/or Dipyridamole Affect Filopodia and Lamellipodia Formation

F-actin stress fibers are major contractile structures that are prominent in fibroblasts, smooth muscle cells, endothelial cells, and some cancer cell lines [22]. In many cancer cells with increased motility, F-actin stress fibers that form longitudinally across the cell body are often observed. At the periphery of the cell membrane, plasma membrane structures called filopodia and lamellipodia are formed, and cell migration occurs via protrusion at the front, mediated by these structures [23,24]. When F-actin was stained with phalloidin in MDA-MB-435 and SK-MEL-5 cells, more stress fibers were observed in MDA-MB-435 cells than in SK-MEL-5 cells (Appendix A). The effects of statins and/or dipyridamole on the formation of filopodia and lamellipodia were found using a scanning electron microscope (SEM), especially in MDA-MB-435 cells (Appendix A).

### 3.5. Atorvastatin Upregulates HMGCR mRNA Expression in a Dose-Dependent Manner While Dipyridamole Tends to Downregulate It

*HMGCR* expression increased upon statin treatment in a dose-dependent manner. However, the level of upregulation depended on the cancer cell type. Moreover, dipyridamole tended to downregulate *HMGCR* expression (Appendix A).

### 3.6. Dipyridamole Augments Atorvastatin’s Anticancer Effect, Which Correlates with Its Attenuation of Statin-Induced Increase in HMGCR Expression

In all five human and canine melanoma cell lines, HMGCR protein expression at the steady state was not detected (extremely low level). The addition of atorvastatin alone drastically increased HMGCR protein expression, whereas the addition of dipyridamole alone did not increase its expression. However, dipyridamole significantly attenuated the statin-induced increase in HMGCR protein levels in all five atorvastatin co-treated cell lines (Figure 2).

### 3.7. Geranylgeranyl Pyrophosphate (GGPP) Rescues Cell Proliferation Inhibited by Dipyridamole Combination with Atorvastatin

The addition of GGPP attenuated the inhibitory effect of atorvastatin–dipyridamole co-treatment and significantly rescued (*p* < 0.01) cell proliferation in all cell lines examined: SK-MEL-5 and MDA-MB-435 (human melanoma), and YCC, ITP, and HTR (canine melanoma) (Figure 3). This also demonstrates that the decrease in the melanoma cell line viability by the statin–dipyridamole co-treatment is an on-target (i.e., HMGCR) effect on the mevalonate pathway.

### 3.8. Vemurafenib Augments the Combined Inhibitory Effect of Statin and Dipyridamole

The MDA-MB-435 and SK-MEL-5 melanoma cells lines are known to harbor BRAF Val600Glu (V600E) mutations and to be sensitive to the BRAF inhibitor vemurafenib [25,26]. Indeed, vemurafenib treatment inhibited cell proliferation in a dose-dependent manner 72 h after the start of drug addition (Figure 4A,B). When used alone at a concentration that exhibited a 20–30% cytostatic effect on cell growth of the two human melanoma cell lines, vemurafenib did not augment the inhibitory effect on cell proliferation with atorvastatin alone, but it did so in the presence of atorvastatin and dipyridamole co-treatment (Figure 4C,D).

## 4. Discussion

Melanoma is curable when diagnosed early in its localized form. In contrast, metastatic melanoma has a very poor prognosis, and new strategies need to be explored by combining conventional chemo- and/or targeted therapies with new drugs to improve therapeutic outcomes [27]. However, Phase II and III clinical trials are expensive, and are a major hurdle to the development of new drugs. In contrast, the development of anticancer drugs through drug repositioning is much cheaper and may have a lower risk of failure [28]. Statins are such potential anticancer agents that target the mevalonate pathway [8].

To date, attempts have been made to augment the tumor-inhibitory effect of statins, which can be classified into three groups from the perspective of statin combination therapy. The first group of compounds includes those that exert their effects by interfering with the statin-induced activation of SREBP2-driven mevalonate pathway enzymes (including HMGCR). This group includes drugs that block the SREBP2-mediated feedback response by blocking SREBP2 processing (e.g., dipyridamole) [29,30] and HMGCR protein degraders (e.g., SR-12813) [20]. The second group includes drugs that likely exert their effects through alternative mechanism(s). Our own expression profile-based study identified BRAF inhibitors (e.g., dabrafenib), MEK inhibitors (e.g., selumetinib), Bcl-2/Bcl-x/Mcl-1 inhibitors (TW37), NF-ĸB inhibitors (piperlongumine), and HSP-90 inhibitors (elesclomol) as candidates for potentiating statins’ inhibitory effects [31]. Other studies have shown additive or synergistic effects of combination therapy with statins, including with BRAF inhibitors [32,33], MEK inhibitors [34], and other inhibitors of the Ras-Raf-MEK-ERK pathway [35]. The third group includes those in which either group 1 and/or 2 may be relevant (e.g., nelfinavir, honokiol, and clotrimazole) [36]. Van Leeuwen et al. [36] expanded the number of agents similar to dipyridamole that augment the anticancer effect of fluvastatin, using computational pharmacogenomics, and suggested a strong need for further investigation of fluvastatin–nelfinavir, fluvastatin–honokiol, fluvastatin–clotrimazole, and fluvastatin–vemurafenib combinations.

In this study, we demonstrated the potential efficacy of the combination of atorvastatin and dipyridamole for the treatment of melanoma. The IC_50_ values of atorvastatin in both human and canine melanoma cell lines showed a 68–92% reduction when combined with dipyridamole compared with those of atorvastatin alone (Table 1 and Appendix A). Notably, in some melanoma cells, proliferation was suppressed to almost zero with the combination treatment (≥3 μM atorvastatin; Figure 1). The concentration of dipyridamole used in this study is close to the plasma levels in patients taking this drug [37,38]. By contrast, given that pharmacologically relevant concentrations of atorvastatin for treatment range from 0.1 to 0.3 μM [39], the 3-μM dose used herein is high; however, unlike in the treatment of dyslipidemia, it may be an acceptable concentration if it inhibits or kills cancer cells aggressively, in a single high dose, rather than being taken continuously. Collisson et al. [17] reported that in vitro, atorvastatin altered the morphology and Rho localization and attenuated RhoC signaling in human melanoma cells; however, in vivo, atorvastatin inhibited the invasion and metastasis of melanoma cells but did not affect their growth. As described above, they demonstrated that atorvastatin specifically inhibited metastasis in vivo rather than cell proliferation and stated that the potential clinical benefit of statin therapy did not correlate with the ability to kill cancer cells themselves. Collisson et al. [17] concluded that statins, at the levels routinely prescribed for hypercholesterolemia, may have specific antimetastatic effects. Based on their report and considering our results, at low concentrations of atorvastatin, which did not inhibit cell proliferation, it is possible that an anti-metastatic effect rather than an anti-cell death effect was in action. In fact, in MDA-MB-435, 5 μM atorvastatin single treatment barely inhibited cell proliferation (Appendix A); however, it affected the formation of filopodia and lamellipodia (Appendix A, upper middle panel). In contrast, they were affected considerably by 5 μM atorvastatin and dipyridamole combination, inhibiting cell proliferation by approximately 80% (Appendix A upper right panel). In SK-MEL-5, 5 μM atorvastatin single treatment slightly affected filopodia and lamellipodia formation (Appendix A lower middle panel). As the atorvastatin concentration of 5 μM was almost the same as the IC_50_ of the combination treatment, cytoplasmic atrophy was evident and almost no filopodia or lamellipodia were observed (Appendix A lower right panel). These results are not unexpected considering that the depletion of FPP and GGPP, the mevalonate pathway intermediates, by statins leads to interference with the post-translational modification and activation of Ras, Rho, or Rac [7,8]. Rho, Cdc42, and Rac regulate stress fibers, filopodia, and lamellipodia, respectively [40]. In our recent study, in which the inhibitory effects of various statins (atorvastatin, pitavastatin, fluvastatin, simvastatin, and rosuvastatin) on the cell proliferation of canine melanoma were investigated, we found that pitavastatin had the strongest inhibitory effect on cell proliferation [15]. If pitavastatin is combined with dipyridamole, lower statin concentrations may inhibit cancer cell proliferation. Additionally, the inhibition of cell proliferation by the combination of dipyridamole and atorvastatin was rescued by supplementation with GGPP, a metabolic intermediate of the mevalonate pathway, which showed that dipyridamole had an on-target effect on the mevalonate pathway (Figure 3). Notably, both statins and dipyridamole target the mevalonate pathway. However, statins induce HMGCR upregulation, whereas dipyridamole augments the antitumor effects of statins without inducing HMGCR upregulation (Figure 2).

The BRAF inhibitor vemurafenib is the standard treatment for unresectable or metastatic melanomas harboring activating BRAF V600E mutations [1]. However, its long-term efficacy is often reduced due to the invariable development of resistance. For melanoma treatment with BRAF inhibitors, metabolic changes, such as increased dependency on lipogenesis, have been suggested as adaptive mechanisms [41]. In addition, it has been reported that statin treatment increases the vulnerability of PGC1α-suppressed BRAF-inhibitor resistant melanomas [42], and it may be effective against melanomas that gradually develop resistance to vemurafenib. Indeed, we have found that vemurafenib further potentiated the effect of the combined statin–dipyridamole treatment on human melanoma cell lines (Figure 4). Wang et al. [43] reported that HMGCR expression was higher in vemurafenib-resistant melanoma cells and that the HMGCR level correlated well with vemurafenib resistance. They demonstrated that combined treatment with vemurafenib and physapubenolide enhanced sensitivity to vemurafenib by decreasing HMGCR expression. Therefore, this result (Figure 4) is medically relevant, and targeting SREBP2-driven mevalonate pathway upregulation may offer a new avenue for treating melanomas resistant to vemurafenib or to potentiate the effect of vemurafenib.

Furthermore, concomitant treatment with FDA-approved dacarbazine and simvastatin or fluvastatin suppressed both tumor growth and metastasis in murine models by attenuating the RhoA/RhoC pathway [44]. In colon cancer cells, dipyridamole augmented the cytotoxicity of the MEK1/2 inhibitor, trametinib, and was used to treat unresectable or metastatic malignant melanomas with BRAF mutations [1] through the dual targeting of the HMGCS1 and MEK pathways [45]. Based on this evidence, statins and dipyridamole are likely to be compatible therapeutic agents for advanced melanoma treatment.

Melanomas occur not only in humans, but also in many animals, including dogs, cats, horses, and pigs; however, they are more common in dogs than in other species [46]. Domestic dogs (*Canis lupus familiaris*) spontaneously develop melanoma and are exposed to environmental factors similar to those experienced by humans; they have a physiology similar to that of humans, indicating that dogs are excellent animal models for the development of novel therapies for melanoma [47]. Thus, comparative oncology, which compares humans with companion animals living in similar or shared environments, can provide useful insights into developmental mechanisms and treatments of human melanoma. Several studies, including ours, have shown that statins exert anticancer effects on canine tumors [16,48,49,50]. Furthermore, atorvastatin is well tolerated and has no apparent adverse effects or biochemical abnormalities in either healthy dogs or dogs with congestive heart failure [51]. Therefore, statins are considered suitable for cancer treatment in dogs. In addition, dipyridamole is more cytotoxic toward cancer cells than normal cells [52]. Given the lack of effective therapies for melanoma in dogs [47], statin–dipyridamole combination therapy may be a good treatment option, and the outcome from therapy in canines can be used as a reference for human melanoma treatment (Figure 5).

One limitation of this study is that the cytotoxicity of atorvastatin on hepatocytes, cardiomyocytes, or skeletal muscle cells was not evaluated. To apply statin therapy in the clinical setting, cytotoxic evaluation in normal cells is essential. Lipophilic statins, such as atorvastatin, lovastatin, cerivastatin, fluvastatin, and atorvastatin, decreased cell viability at increasing concentrations in human cardiomyocytes, murine skeletal muscle cells, cultured human hepatocytes, and primary rat hepatocytes, while hydrophilic statins (rosuvastatin and pravastatin) did not affect the viability [53,54,55]. In a rat skeletal muscle cell line, 100 μM atorvastatin showed cytotoxicity, while pravastatin (up to 1 mM) did not [56]. According to these reports, the concentrations of atorvastatin used this study (3 or 10 μM) do not seem to affect normal cells; however, it may be necessary to examine the efficacy of less cytotoxic hydrophilic statin and dipyridamole combination treatments in cancer cells.

## 5. Conclusions

Dipyridamole is a suitable potentiator of statins, and the following may be conceivable in statin–dipyridamole therapy for advanced melanoma: (1) the attenuation of metastasis formation with a well-tolerated statin-dipyridamole combination that can be safely administered on a long-term basis (with potential use as a postoperative adjuvant treatment), (2) the dual inhibition of distinct pathways compared to conventional therapeutic agents for melanoma, and (3) the use of statin–dipyridamole combination therapy in cases where conventional therapeutic agents cannot be applied.

## Figures and Tables

**Figure 1 biomedicines-12-00698-f001:**
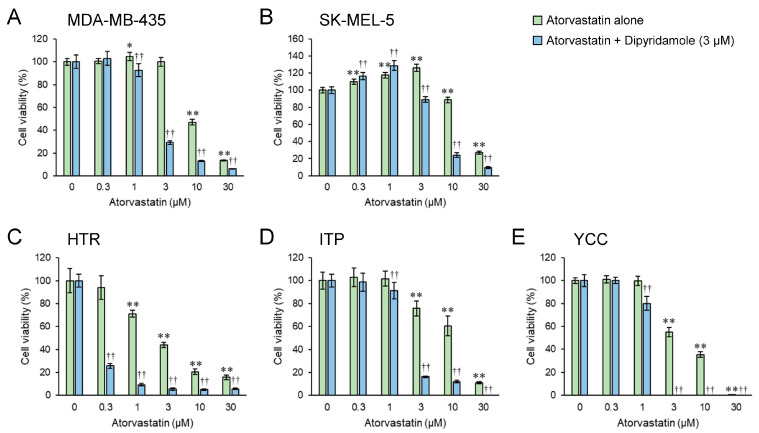
Enhancement of the growth inhibitory effect of atorvastatin on tumor cell lines by combination treatment with dipyridamole. Relative growth of human MDA-MB-435 (**A**) and SK-MEL-5 (**B**) and canine HTR (**C**), ITP (**D**), and YCC (**E**) melanoma cell lines is shown. Green and blue bars represent atorvastatin single treatment group and atorvastatin–dipyridamole combination treatment group, respectively. Values in DMSO control were set at 100%. Each value represents the mean ± SD (n = 3). Data were analyzed using a one-way analysis of variance (ANOVA) with Dunnett’s test for multiple group comparisons. * *p* < 0.05, ** *p* < 0.01, ^††^
*p* < 0.01 with respect to each control group.

**Figure 2 biomedicines-12-00698-f002:**
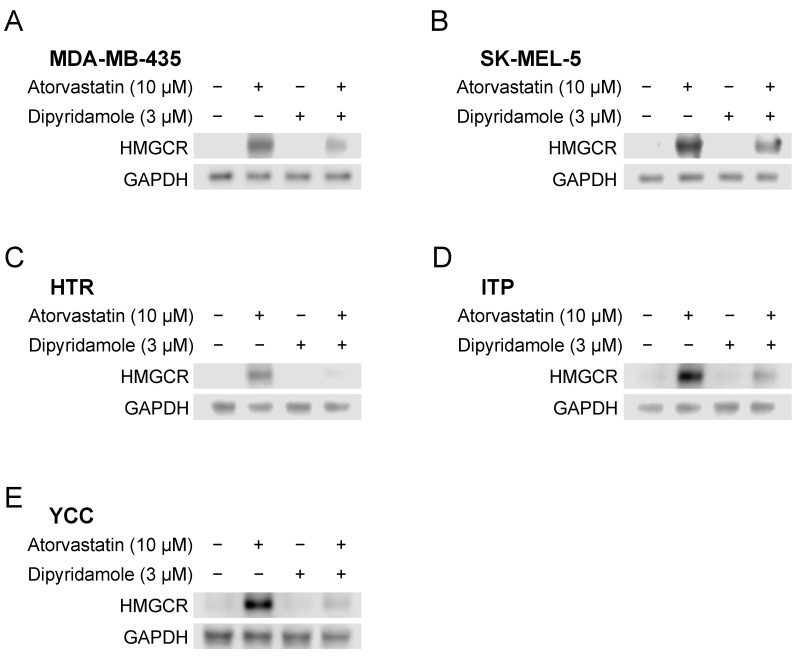
HMGCR expression after atorvastatin, dipyridamole, and their combination treatments. HMGCR protein expression in MDA-MB-435 (**A**), SK-MEL-5 (**B**), HTR (**C**), ITP (**D**), and YCC (**E**) cells was determined using Western blotting. From the left side: lane 1, vehicle control group; lane 2, 10 μM atorvastatin treatment group; lane 3, 3 μM dipyridamole treatment group; and lane 4, 10 μM atorvastatin plus 3 μM dipyridamole treatment group. The Western blot shown is representative of experiments performed in triplicate.

**Figure 3 biomedicines-12-00698-f003:**
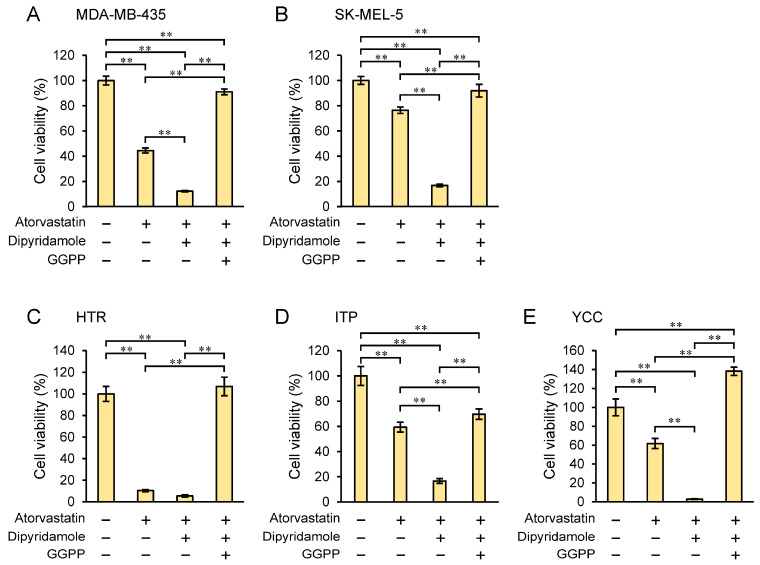
Rescue of atorvastatin- and/or dipyridamole-induced growth inhibition by GGPP. Relative growth of human (MDA-MB-435; (**A**) and SK-MEL-5; (**B**)) and canine (HTR; (**C**), ITP; (**D**), and YCC; (**E**)) melanoma cell lines in the presence or absence of exogenous geranylgeranyl pyrophosphate (GGPP). Values in vehicle control were set to 100%. Each value represents the mean ± SD (n = 3). Data were analyzed using one-way analysis of variance (ANOVA) with the Tukey–Kramer post hoc test for multiple group comparisons. ** *p* < 0.01.

**Figure 4 biomedicines-12-00698-f004:**
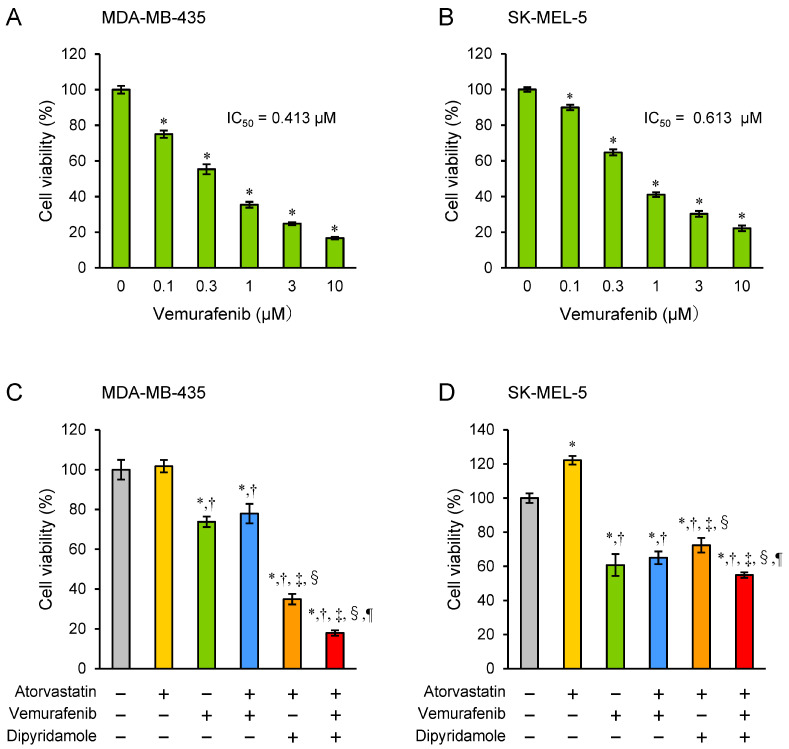
Enhancement of growth inhibitory effect of atorvastatin–dipyridamole combination on tumor cell lines by vemurafenib co-treatment. The relative growth of human melanoma cell lines 72 h after the start of drug treatment is shown for (**A**) vemurafenib single treatment of MDA-MB-435 cells and (**B**) vemurafenib single treatment of SK-MEL-5 cells, at the indicated drug concentrations. (**C**) Single or combined treatment with 0.1 µM vemurafenib, 3 µM atorvastatin, and 3 µM dipyridamole of MDA-MB-435 cells, and (**D**) single or combined treatment with 0.3 µM vemurafenib, 3 µM atorvastatin, and 3 µM dipyridamole of SK-MEL-5 cells at the 72 h time points are shown. Values in DMSO control were set at 100%. Each value represents the mean ± SD (n = 3). (**A**,**B**) Data were analyzed using one-way analysis of variance (ANOVA) with Dunnett’s test for multiple group comparisons. * *p* < 0.01 with respect to each control group. (**C**,**D**) Control, atorvastatin, vemurafenib, atorvastatin+vemurafenib, atorvastatin+dipyridamole, and atorvastatin+vemurafenib+dipyridamole were indicated by grey, yellow, green, blue, orange, and red bars, respectively. Data were analyzed using one-way analysis of variance (ANOVA) with the Tukey–Kramer post hoc test for multiple group comparisons. * *p* < 0.01 (vs. control), ^†^
*p* < 0.01 (vs. atorvastatin), ^‡^
*p* < 0.01 (vs. vemurafenib), ^§^
*p* < 0.01 (vs. atorvastatin + vemurafenib), ^¶^
*p* < 0.01 (vs. atorvastatin + dipyridamole).

**Figure 5 biomedicines-12-00698-f005:**
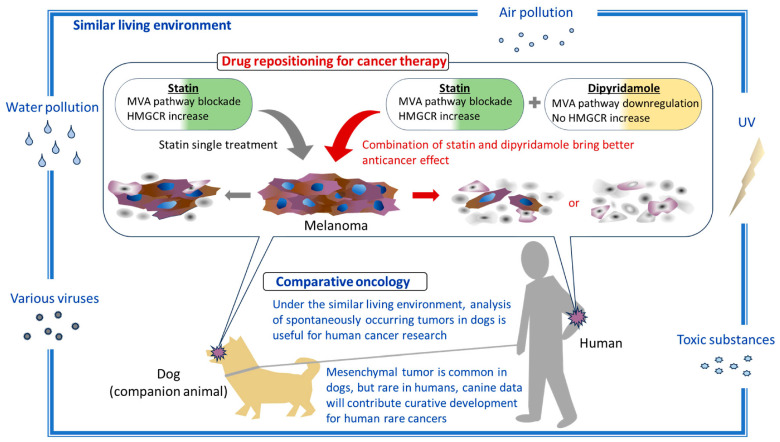
Comparative oncology: overview. The potential of the combination of statins and dipyridamole for cancer treatment through drug repositioning and the potential of comparative oncology in the analysis of spontaneously occurring cancers in domestic dogs that share the same living environment as humans are depicted. MVA pathway; mevalonate pathway.

**Table 1 biomedicines-12-00698-t001:** IC_50_ of atorvastatin.

Melanoma	IC_50_ Atorvastatin (μM)	IC_50_ Combination ^¶^ (μM)	Reduction (%)
MDA-MB-435	9.583	2.180	77
SK-MEL-5	21.769	5.618	74
HTR	2.286	0.184	92
ITP	14.466	1.715	88
YCC	3.507	1.117	68

^¶^: IC_50_ of atorvastatin in the combination.

## Data Availability

All relevant data supporting the key findings of this study are available within the article and its Supplementary Information files or from the corresponding author on reasonable request.

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
