# Peer review of "Repurposing of the Cardiovascular Drug Statin for the Treatment of Cancers: Efficacy of Statin–Dipyridamole Combination Treatment in Melanoma Cell Lines"

_biomedicines, 2024, doi:10.3390/biomedicines12030698_

Round 1

Reviewer 1 Report

Comments and Suggestions for Authors

Comments:

When talking about gene expression gene symbols should be presented in italics

Lines 90, 91 – it is not clear “have been reported to inhibit xenograft metastasis of human melanoma cells overexpressing RhoC, which drastically increases the metastatic potential of melanoma”, isn’t contradictory?

 Results and discussion – it seems that both dipyridamole and atorvastatin produce inhibitory effects in microM concentrations, that are much higher than those reached in clinical conditions; it should be clearly stated and commented; statins are discussed but not dypiridamole

Figure 1 legend; what the colors mean, green and blue ? green (atorvastatin) blue (combination) ? the information should be provided in the figure legend

 There are missing data on dipyridamole alone ??  (if the above is true)

 Figure 2 - HMGCR protein expression bands – it seems that dipyridamole is more potent in suppression than the combination and atorvastatin alone. Atorvastatin attenuates the activity of dipyridamole ??

Suggestions:

Do you have data on hepatocyte or skeletal muscle cells in vitro toxicity of atorvastatin used at the study concentrations?

Author Response

Responses to the reviewers’ comments

We thank the reviewers for reviewing our manuscript and for providing valuable suggestions to improve it. Our responses to the comments are provided below. We have rechecked and revised the entire manuscript, and we have highlighted only those parts that were revised in response to the reviewers’ comments in red font.

Reviewer #1
Comments:

When talking about gene expression gene symbols should be presented in italics.

Response: We thank you for pointing this out. Per your suggestion, we have revised the manuscript as follows:

Original: HMGCR activation (Line 179)

Revised: HMGCR expression

Original: HMGCR mRNA (Line 458)

Revised: HMGCR mRNA

Lines 90, 91 – it is not clear “have been reported to inhibit xenograft metastasis of human melanoma cells overexpressing RhoC, which drastically increases the metastatic potential of melanoma”, isn’t contradictory?

Response: We thank you for the comment. We have revised the relevant part for clarity (Lines 110–111).

Original: …, have been reported to inhibit xenograft metastasis of human melanoma cells overexpressing RhoC, which drastically increases the metastatic potential of melanoma (Collisson et al., 2003).

Revised: …, have been reported to inhibit metastasis of human melanoma cells overexpressing RhoC in a xenograft model (Collisson et al., 2003).

Results and discussion – it seems that both dipyridamole and atorvastatin produce inhibitory effects in microM concentrations, that are much higher than those reached in clinical conditions; it should be clearly stated and commented; statins are discussed but not dipyridamole.

Response: We thank you for the constructive comment. We have added the description of dipyridamole (Lines 244–246). Also, we have added references (Line 531–533, 581–584).

Original: …(≥3 μM atorvastatin; Fig. 1). Given that…

Revised: …(≥3 μM atorvastatin; Fig. 1). The concentration of dipyridamole used in this study is close to the plasma levels in patients taking this drug (Guo et al., 2010; Liu et al., 2004). By contrast, given that…

References

Guo, S., Stins, M., Ning, M., & Lo, E. H. (2010). Amelioration of inflammation and cytotoxicity by dipyridamole in brain endothelial cells. Cerebrovascular Diseases, 30(3), 290–296. https://doi.org/10.1159/000319072

Liu, Y., Cone, J., Le, S. N., Fong, M., Tao, L., Shoaf, S. E., …Sun, B. (2004). Cilostazol and Dipyridamole Synergistically Inhibit Human Platelet Aggregation. Journal of Cardiovascular Pharmacology, 44(2), 266–273. https://doi.org/10.1097/00005344-200408000-00017

Figure 1 legend; what the colors mean, green and blue ? green (atorvastatin) blue (combination) ? the information should be provided in the figure legend.

There are missing data on dipyridamole alone ??  (if the above is true)

Response: We appreciate the comment. We have added the explanation for the colors in the figure legend (Line 704–706). The data on dipyridamole alone are in Supplementary Figure S2.

Original: …melanoma cell lines is shown. Values in DMSO control were…

Revised: …melanoma cell lines is shown. Green and blue bars represent atorvastatin single treatment group and atorvastatin-dipyridamole combination treatment group, respectively. Values in DMSO control were…

Figure 2 - HMGCR protein expression bands – it seems that dipyridamole is more potent in suppression than the combination and atorvastatin alone. Atorvastatin attenuates the activity of dipyridamole ??

Response: We thank you for the comment. HMGCR expression level at steady state is extremely low (Atorvastatin −, Dipyridamole −). Atorvastatin addition increases HMGCR expression drastically (Atorvastatin +, Dipyridamole −); however, dipyridamole addition does not induce HMGCR expression (Atorvastatin −, Dipyridamole +). When atorvastatin and dipyridamole are combined, dipyridamole attenuates the HMGCR expression indued by atorvastatin (Atorvastatin +, Dipyridamole +). Therefore, to avoid confusion, we have added the description of HMGCR expression level at the steady state in the Result section (Lines 183–184).

Original: In all five human and canine melanoma cell lines, the addition of atorvastatin alone drastically increased HMGCR protein expression, whereas the addition of dipyridamole alone did not increase its expression.

Revised: In all five human and canine melanoma cell lines, HMGCR protein expression at the steady state was not detected (extremely low level). The addition of atorvastatin alone drastically increased HMGCR protein expression, whereas the addition of dipyridamole alone did not increase its expression.

Suggestions:

Do you have data on hepatocyte or skeletal muscle cells in vitro toxicity of atorvastatin used at the study concentrations?

Response: We appreciate your constructive suggestion. We do not have data on cytotoxicity, but there have been papers evaluating the cytotoxicity of atorvastatin in hepatocytes and skeletal muscle cells in vitro. Those data will be important information for readers; therefore, we have added the description about that as a limitation of this study before the conclusion as follows (Lines 329–340):

One limitation of this study is that the cytotoxicity of atorvastatin on hepatocytes, cardiomyocytes, or skeletal muscle cells was not evaluated. To apply statin therapy in the clinical setting, cytotoxic evaluation in normal cells is essential. Lipophilic statins, such as atorvastatin, lovastatin, cerivastatin, fluvastatin, and atorvastatin, decreased cell viability at increasing concentrations in human cardiomyocytes, murine skeletal muscle cells, cultured human hepatocytes, and primary rat hepatocytes, while hydrophilic statins (rosuvastatin and pravastatin) did not affect the viability (Kubota et al., 2004; Shu et al., 2016; Zhang et al., 2022). In a rat skeletal muscle cell line, 100 μM atorvastatin showed cytotoxicity, while pravastatin (up to 1 mM) did not (Kaufmann et al., 2006). According to these reports, the concentrations of atorvastatin used this study (3 or 10 μM) seems not to affect normal cells; however, it may be necessary to examine the efficacy of less cytotoxic hydrophilic statin and dipyridamole combination treatment in cancer cells.

With the description of the limitation, we have added the following references (Lines 553–556, 564–567, 623–626, 687–690):

References

Kaufmann, P., Török, M., Zahno, A., Waldhauser, K. M., Brecht, K., & Krähenbühlet S. (2006). Toxicity of statins on rat skeletal muscle mitochondria. Cellular and Molecular Life Sciences, 63, 2415–2425. https://doi.org/10.1007/s00018-006-6235-z

Kubota, T., Fujisaki, K., Itoh, Y., Yano, T., Sendo, T., & Oishi, R. (2004). Apoptotic injury in cultured human hepatocytes induced by HMG-CoA reductase inhibitors. Biochemical Pharmacology, 67(12), 2175–2186. https://doi.org/10.1016/j.bcp.2004.02.037

Shu, N., Hu, M., Ling, Z., Liu, P., Wang, F., Xu P., …Liu, L. (2016). The enhanced atorvastatin hepatotoxicity in diabetic rats was partly attributed to the upregulated hepatic Cyp3a and SLCO1B1. Scientific Reports 6, 33072. https://doi.org/10.1038/srep33072

Zhang, Q., Qu, H., Chen, Y., Luo, X., Chen, C., Xiao, B, … Yu, Y. (2022). Atorvastatin induces mitochondria-dependent ferroptosis via the modulation of Nrf2-xCT/GPx4 axis. Frontiers in Cell and Developmental Biology, 10, 806081. https://doi.org/10.3389/fcell.2022.806081

Reviewer 2 Report

Comments and Suggestions for Authors

The manuscript entitled, “Repurposing of cardiovascular-drug statin for treatment of cancers: efficacy of statin-dipyridamole combination treatment in melanoma cell lines” is interesting. The authors determined the efficacy of statin-dipyridamole combination treatment using human and spontaneously occurring canine melanoma cell lines. Overall, the data demonstrated that compared to atorvastatin alone, its combination with dipyridamole inhibits cell proliferation at a greater rate, and that this effect was enhanced by the combination of BRAF kinase inhibitor, vemurafenib, in BRAF V600E mutation harboring melanoma cell lines. The studies are nicely designed and executed. However, there are a few comments that need to be addressed.

Major comments:

 1.     The western blot images of only experiment, but not all 3 experiments were provided. 

 Minor comments:

 1.    The Figure 4C, D data demonstrate that vemurafenib did not augment the inhibitory effect on cell proliferation with atorvastatin alone, but it did so in the presence of atorvastatin and dipyridamole co-treatment. However, the reason for this effect/discrepancy is not discussed.

2    2.  In the Statistical analysis section, please mention that all the experiments were repeated at least three times independently.

Author Response

Responses to the reviewers’ comments

We thank the reviewers for reviewing our manuscript and for providing valuable suggestions to improve it. Our responses to the comments are provided below. We have rechecked and revised the entire manuscript, and we have highlighted only those parts that were revised in response to the reviewers’ comments in red font.

Reviewer #2

Comments and Suggestions for Authors:

The manuscript entitled, “Repurposing of cardiovascular-drug statin for treatment of cancers: efficacy of statin-dipyridamole combination treatment in melanoma cell lines” is interesting. The authors determined the efficacy of statin-dipyridamole combination treatment using human and spontaneously occurring canine melanoma cell lines. Overall, the data demonstrated that compared to atorvastatin alone, its combination with dipyridamole inhibits cell proliferation at a greater rate, and that this effect was enhanced by the combination of BRAF kinase inhibitor, vemurafenib, in BRAF V600E mutation harboring melanoma cell lines. The studies are nicely designed and executed. However, there are a few comments that need to be addressed.

Response: We would like to thank the reviewer for the valuable comments and insightful suggestions. We hope that our point-by-point responses have satisfactorily addressed your concerns.

Major comments:

  1. The western blot images of only experiment, but not all 3 experiments were provided.

Response: We thank you for pointing out the insufficient description of the western blot data. We have added the following sentences in the figure legend (Lines 717–718).

The western blot shown is representative of experiments performed in triplicate.

Minor comments:

  1. The Figure 4C, D data demonstrate that vemurafenib did not augment the inhibitory effect on cell proliferation with atorvastatin alone, but it did so in the presence of atorvastatin and dipyridamole co-treatment. However, the reason for this effect/discrepancy is not discussed.

Response: We appreciate your constructive suggestion. Per your suggestion, we have revised the text follows (Lines 294–298). Also, we have added a reference (Line 667–670).

Original: …statin-dipyridamole treatment on human melanoma cell lines (Fig. 4). Therefore, this result is medically relevant, …

Revised: …statin-dipyridamole treatment on human melanoma cell lines (Fig. 4). Wang et al. (2022) reported that HMGCR expression was higher in vemurafenib-resistant melanoma cells and that HMGCR level is well correlated with vemurafenib resistance. They demonstrated that combined treatment with vemurafenib and physapubenolide enhanced sensitivity to vemurafenib by decreasing HMGCR expression. Therefore, this result is medically relevant, …

Reference

Wang, H.Y., Yu, P., Chen, X.S., Wei, H., Cao, S.J., Zhang, M., … Cheng, Y. (2022). Identification of HMGCR as the anticancer target of physapubenolide against melanoma cells by in silico target prediction. Acta Pharmacologica Sinica, 43, 1594–1604. https://doi.org/10.1038/s41401-021-00745-x

  1. In the Statistical analysis section, please mention that all the experiments were repeated at least three times independently.

Response: We thank you for pointing this out. We have added the following sentence in the last part of the Statistical analysis section (Lines 460–461).

All the experiments were repeated at least three times independently.

Reviewer 3 Report

Comments and Suggestions for Authors

This is an interesting manuscript. I have no major comments/suggestions except those concerning the Introduction. Namely, in Introduction the authors should mention that statins have anticancer effects in different types of cancer (lung, kidney, bladder, colon etc.) and the references might be: Lashgari NA, et al.Statins block mammalian target of rapamycin pathway: a possible novel therapeutic strategy for inflammatory, malignant and neurodegenerative diseases. Inflammopharmacology. 2023 Feb;31(1):57-75.

AND

Amin F, et al.The role of statins in lung cancer.

Arch Med Sci. 2021 Mar 18;18(1):141-152.

Comments on the Quality of English Language

English is quite good.

Author Response

Responses to the reviewers’ comments

We thank the reviewers for reviewing our manuscript and for providing valuable suggestions to improve it. Our responses to the comments are provided below. We have rechecked and revised the entire manuscript, and we have highlighted only those parts that were revised in response to the reviewers’ comments in red font.

Reviewer #3
Comments and Suggestions for Authors:

This is an interesting manuscript. I have no major comments/suggestions except those concerning the Introduction. Namely, in Introduction the authors should mention that statins have anticancer effects in different types of cancer (lung, kidney, bladder, colon etc.) and the references might be: Lashgari NA, et al. Statins block mammalian target of rapamycin pathway: a possible novel therapeutic strategy for inflammatory, malignant and neurodegenerative diseases. Inflammopharmacology. 2023 Feb;31(1):57-75. AND Amin F, et al. The role of statins in lung cancer. Arch Med Sci. 2021 Mar 18;18(1):141-152.

Response: We sincerely appreciate the positive feedback. Per your suggestion, we have mentioned that statins have anticancer effects in different types of cancer (lung, kidney, bladder, colon, breast, prostate etc.) in the Introduction section (Line 90–93) and have added the following references (Line 499–501, 568–572, 638–641, 684–686).

References

Amin, F., Fathi, F., Reiner, Ž., Banach, M., & Sahebkar, A. (2021). The role of statins in lung cancer. Archives of Medical Science, 18(1), 141–152. https://doi.org/10.5114/aoms/123225

Lashgari, N. A., Roudsari, N. M., Zadeh, S. S. T., Momtaz, S., Abbasifard, M., Reiner, Ž., … Sahebkar. A. (2023). Statins block mammalian target of rapamycin pathway: a possible novel therapeutic strategy for inflammatory, malignant and neurodegenerative diseases. Inflammopharmacology, 31, 57–75. https://doi.org/10.1007/s10787-022-01077-w

Tilija Pun, N., & Jeong, C.-H. (2021). Statin as a Potential Chemotherapeutic Agent: Current Updates as a Monotherapy, Combination Therapy, and Treatment for Anti-Cancer Drug Resistance. Pharmaceuticals, 14(5), 470. https://doi.org/10.3390/ph14050470

Zaky, M. Y., Fan, C., Zhang, H., & Sun, X. -F. (2023). Unraveling the Anticancer Potential of Statins: Mechanisms and Clinical Significance. Cancers, 15(19), 4787. https://doi.org/10.3390/cancers15194787

Round 2

Reviewer 1 Report

Comments and Suggestions for Authors

I would suggest to accept the manuscript in the present form